# Impact of resource scarcity on general categorization tendency: The moderating role of perceived mutability

**Haowen Xiao**, **Jiayi Guo, Junyi Fang, Ting Xiao, Guocheng Li***

Management School, Hainan University, Haikou, Hainan, China

* 1154602066@qq.com

**Data Availability Statement:** All relevant data are within the paper and its Supporting Information files.

**Funding:** This research was supported by Hainan Provincial Natural Science Foundation of China

## Abstract

Prior studies revealed varying effects of resource scarcity on individuals' general categorization tendency. However, little is known about when and why such differences occur. Based on the self-regulatory model of resource scarcity, we examine whether resource scarcity generates higher or lower general categorization tendency depending on the perceived mutability of the resource discrepancy. We conducted two online experiments to test the hypotheses. The results affirmed that when individuals consider the resource discrepancy to be mutable, they are more likely to seek abundance to compensate for resource scarcity, thus reducing their general categorization tendency. In contrast, perceiving the scarcity as immutable triggers the intention to restore a sense of control undermined by the scarcity, increasing individuals' general categorization tendency. Our findings provide insights into the downstream consequences of resource scarcity and offer significant managerial implications for coping strategies.

## Introduction

Individuals usually perceive resources, including money [1], time [2], and products [3] as scarce. Prior literature indicated that resource scarcity is a "mindset" which implies "having less than you want" [4]. It can significantly impact individual perceptions, motivations, judgments, and choices [2, 5–11]. Recent studies examined how the general sense of resource scarcity influences individuals' general categorization tendency, which refers to a human's basic motive to seek structures and order by the process of dividing the world into manageable groups of entities [9, 12, 13]

However, past literature provided contradictory results regarding the impact of resource scarcity on subsequent general categorization tendency. One stream of literature demonstrated that resource scarcity increases individuals' general tendency to categorize objects, as doing so could provide them with the opportunity to restore their sense of control, which is degraded by the resource scarcity [10, 14, 15]. Another stream of research indicated that resource scarcity reduces individuals' general categorization tendency by motivating them to seek abundance to compensate for the scarcity [9, 16]. Therefore, it is reasonable to assume that there are specific conditions wherein resource constraints may either cause a high or low categorization tendency.

(Grant No. 722QN291, 623RC453, 723QN215). The funders had no role in study design, data collection and analysis, decision to publish, or preparation of the manuscript.

**Competing interests:** The authors have declared that no competing interests exist.

This study employs the self-regulatory model of resource scarcity to establish that an unfavorable resource scarcity-related discrepancy may increase or decrease one's general categorization tendency depending on the perceived mutability of the resource discrepancy. It also explores the underlying psychological mechanisms of these effects. We argue that individuals adjust their behavior to alleviate the aversive discrepancy observed in resource scarcity depending on the degree of perceived mutability, that is, the extent to which a person feels they may alter the scarcity discrepancy [14, 17]. Thus, we believe that after experiencing resource scarcity, people assess the possibility of reducing the observed differences that affect their way of coping with resource scarcity. Specifically, when those who have experienced resource scarcity infer that they cannot diminish the resource discrepancy despite a decent amount of effort, their sense of control may lessen, leading to a higher general categorization tendency [13, 18]. On the contrary, when individuals believe that putting in an acceptable amount of effort can decrease the resource discrepancy, a desire for abundance is induced, decreasing the general categorization tendency. This is because categorization might lead to the psychological perception that there are fewer accessible things [9].

This study provides a theoretical understanding of several research areas. First, it identifies perceived mutability as a novel boundary condition to reconcile the incongruent predictions about individuals' subsequent general categorization tendency when faced with resource scarcity. Second, it illuminates the psychological mechanisms underlying the varying effects of resource constraints on individuals' general categorization tendency. More specifically, it contributes to existing literature by identifying the desire for abundance and control as mediators of this effect [9, 10, 12]. Although prior research has examined these processes separately, to the best of our knowledge, ours is the first study in which these are converged to explain the impact on individuals' general categorization propensity. Finally, our research extends the self-regulatory model of resource constraints to categorization literature. Previous research on this model and the growing literature on resource scarcity primarily focused on planning and setting priorities [11, 19] and seeking unique items [1], which may improve or impair self-regulatory performance.

## Theoretical background and hypotheses development

### Resource scarcity and general categorization tendency

Resource scarcity is when one perceives or observes a discrepancy between one's existing resources and a more ideal reference [14]. Individuals frequently experience a shortage of resources in their daily lives [3, 20], for instance, in terms of money [12, 21], products [11, 22, 23], food [11, 24, 25], and time [2, 19, 26]. Recent literature showed that a generalized mindset can be activated by the perception of resource scarcity, which affects individual motivations, perceptions, and behavior [5, 6, 19, 22]. For example, resource constraints can enhance an individual's subjective arousal, leading to the polarization of their preferences [11]. Similarly, Shah et al. (2012) suggested that people's attention-allocation patterns are altered by resource scarcity, i.e., they become more invested in some issues while ignoring others. Moreover, simply exposing individuals to advertising information that implies that the desired product is limited in supply might cause them to behave more aggressively, such as choosing more violent video games and acting violently when a vending machine gets jammed [22].

Although many studies investigated how resource scarcity affects individual behavior, they yielded contradictory results about the impact of resource scarcity on general categorization tendency [5, 9, 12–13]. Specifically, about how scarcity cues affect the general inclination toward categorization, which focuses on how much people categorize [9].

More specifically, in response to resource constraints, people may display either an increased or decreased categorization tendency. One stream of the literature demonstrated

that resource scarcity may increase categorization tendency [12, 15, 27], which might help people regain personal control over their lives after feeling degraded by the perception of resource scarcity [10, 14]. In contrast, other studies indicated that resource scarcity may reduce assortment tendency because it promotes motivation to seek more resources [1, 2, 5, 28], as opposed to categorization, which creates a sense of diminution [9]. For instance, a study showed that in a game with numerous rounds, participants with a restricted budget were more likely than those with an adequate budget to borrow greater resources for their present use from later rounds [3]. Furthermore, resource constraints can generate a competitive mindset and lead individuals to make decisions that may increase their well-being [28].

Thus, although previous literature suggested that people's general categorization tendency may differ in the face of resource scarcity, the "when" and "why" remain underexplored. Consequently, a better understanding of the psychological mechanisms engaged in coping with resource constraints and how these affect subsequent categorization tendencies is needed. This study illuminates two different psychological routes by which people cope with resource constraints based on the existing literature on resource constraints and the self-regulatory model of resource constraints [14].

## The moderation role of mutability

As per the self-regulatory model of resource scarcity of Cannon et al. (2019), different levels of mutability can lead to various behavioral outcomes when people are reminded of resource scarcity. Perceived mutability is defined as the extent to which an individual deems that the scarcity condition can be altered by continuing to invest effort, such as time, money, and physical or mental exertion [17]. Specifically, in a low-mutability scenario, individuals believe that even if they invest rational efforts into it, they would be unable to reduce the resource difference; therefore, they attempt to restore personal control [13]. Conversely, in highly-mutable situations, individuals believe that personal investment can help reduce resource scarcity.

Thus, they frequently act in a scarcity-decreasing manner by directly addressing the disparity within an identical domain [14]. In light of the theoretical role of fallibility in coping with resource constraints, individuals' response to resource scarcity relies on how mutable they believe is the resource discrepancy.

It could be possible that perceived mutability plays a moderating role when a general sense of scarcity is triggered. Thus, we posit that individuals' general categorization tendency decreases when perceived mutability is higher because they will be more likely to immediately resolve the resource discrepancy. This is because categorization evokes a feeling of reduction (i.e., many items are divided into smaller subgroups by categorization). However, when perceived mutability is low, individuals perceive the lack of resources as a psychological threat to their sense of control. To compensate for such reduced personal control, an individual is motivated to augment the general categorization tendency in subsequent contexts. Categorizing offers a more ordered and consistent reality, which is a way to create sense of control [29]. Thus, we propose the following hypotheses:

**H1:** Perceived mutability moderates the effect of resource scarcity on general categorization tendency.

**H1a:** When perceived mutability is high, the reminders of resource scarcity lead individuals to decrease their general categorization tendency.

**H1b:** When perceived mutability is low, the reminders of resource scarcity lead individuals to increase their general categorization tendency.

## The mediating role of desire for control

An innate motivation or demand to directly exert control over one's surroundings to achieve desired results or avoid undesired outcomes is called a desire for control [30–32], and research on diverse types of resource scarcity repeatedly showed that people's sense of control may be degraded by perceived resource constraints [18, 21, 33, 34]. For instance, people from financially-restricted families experience much less personal control when confronted with an immediate threat than those from generally high-income families [35]. Moreover, a lower social class (or socioeconomic status) is linked to fewer resources, more threat exposure, and a weakened sense of personal control [34]. Conversely, those with abundant resources feel more in charge and unconstrained, and their daily lives are comparatively protected from external pressures and danger [33].

A lack of personal control engenders an aversive feeling that drives individuals to regain their perception of control to baseline levels [36]. Previous research demonstrated that people who lack personal control are likely to affirm structured interpretations of the world [37, 38]. For instance, research showed that low personal control led to increased subsequent beliefs in the existence of a controlling God [37]. In marketing domain, consumers who lack control tend to seek structure through consumption. Lembregts and Pandelaere (2019) show that low personal control leads consumers to prefer and rely more on numerical attributes as a point value, relative to a range format. This occurs because product attributes specified in a point value format strengthen consumers' belief that the environment is predictable [39].

More importantly, Heider (1958) believed that the underlying motivation driving the cognitive processes involved in perceiving the social world is an effort to create a structure and order in an excessively complicated stimuli environment [40]. Thus, categorization is one method for producing such a sense of structure and order [9, 41]; it is the process by which people divide the world into smaller pieces to make it more manageable and meaningful and to make it more compatible with their unique capacities [42]. Therefore, seeking a greater general categorization tendency may be a useful tactic for regaining the threatened personal control. As noted above, we propose that under conditions of low mutability, when a person perceives an inability to alleviate the discrepancy in resource levels, resource scarcity undermines their feelings of personal control. Thus, individuals who are subjected to resource scarcity show a higher general categorization tendency to restore diminished personal control. More formally,

H2a: When perceived mutability is low, the desire for control mediates the effect of resource scarcity on general categorization tendency.

## The mediating role of desire for abundance

Resource scarcity entails a person's present level of resources being lower than their desired level [14]. Aversive resource scarcity makes individuals pursue abundance and make up for the lack [9, 16]. This is a proposition that has been supported across a range of behavioral domains. For example, financial scarcity prompts consumers to seek more scarce objects that are inaccessible to other buyers in their surroundings [1]. Similarly, Kapoor and Tripathi (2020) suggested that people who experience downward timekeeping are more likely to perceive themselves as resource-scarce, leading to a higher preference for high calorie foods. People become more promotion-oriented when informed of resource scarcity, which raises their expectation of potential better-than-reference outcomes and results in greater preference for range promotion options [5]. Moreover, a general sense of resource constraints triggers a competitive mindset, which engenders individuals to prioritize their own interests over those of

others [28]. These observations imply that the scarcity of various resources, including money, time, goods, or general perceptions, activates the motivation to make up for the resource scarcity and, consequently, the pursuance of the plenty [9].

However, categorization creates a sense of reduction by making people believe that there are fewer available things (for example, dividing 15 cups into 3 groups depending on their small, medium, and big water capacity; [9]). Once this categorization occurs, individuals use the category to draw inferences about and enumerate the objects [43]. People have a subjective inclination to intuitively classify items, and the number of generated categories (e.g., 3) is less than the number of individual items (e.g., 15); thus, this general categorizing tendency may create the perception that there are only few accessible objects [9, 44]. Hence, we propose that under conditions of high mutability, resource scarcity lessens people's general inclination toward categorization because the desire for plenty, brought on by the scarcity, creates a sensation of restriction associated with categorization, which is particularly detestable [2, 16]. Thus, we propose the following hypotheses:

H2b: When perceived mutability is high, the desire for abundance mediates the effect of resource scarcity on general categorization inclination.

To provide evidence for the phenomena we identified and their underlying mechanisms, two experiments were performed using various operationalizations of both resource scarcity and general categorization inclination. Study One established the main effect: in high-mutability situations, a generalized feeling of resource deprivation can trigger a decrease in individuals' general categorization tendency (H1a). However, in low-mutability situations, a general perception of resource shortage can eventually increase individuals' general categorization tendency (H1b). By investigating the mediating role of the desire for control and for abundance, Study Two offers evidence in favor of the mechanism behind the main effect (H2a and H2b).

## Ethical concern

Ethical review and approval were not required for the study on human participants in accordance with the local legislation and institutional requirements. The participants provided their written informed consent to participate in this study.

## Study 1

Study 1 investigates how the perceived mutability of resource discrepancy influences individuals' general categorization tendency (Hypothesis 1). The study participants were instructed about a writing task, which we used to manipulate the resource constraints [28]. Subsequently, four mutability items were completed by the participants (e.g., "With a reasonable amount of effort, I can obtain more resources;" $\alpha = 0.85$), based on the work of Leary et al. (2022). We expected that, compared with the baseline condition, resource scarcity would reduce general categorization tendency in case of perceived mutable scarcity and increase general categorization tendency in case of perceived immutable scarcity.

### Method

**Participants and design.**  A total of 201 participants (63% female; $M_{age}$ = 28.63 years) were recruited on the Credemo platform (a small cash incentive was offered). One of the two resource conditions was randomly assigned to them (resource scarcity vs. baseline).

**Procedure.**  All participants were randomly assigned to one of the two groups. The manipulation procedure described by Roux et al. (2015) was used. First, they completed an episodic recall task. Specifically, in the scenario of scarce resources, the participants were asked to recall

and describe, in detail, a time when they experienced limited resources. In the control group, the participants wrote down what they had done in the past week.

After finishing the recall task, four mutability items were also completed by the participants (e.g., "I can get the resources I need by putting in effort"; $\alpha$ = 0.85); these were drawn from Leary et al. (2022). We assessed general categorization tendency, following Park et al. (2020), using a three-item scale (e.g., "It feels easy for me to see similarities between things"). These questions were rated on a seven-point scale (1 = "not at all," and 7 = "very much"). To obtain the general categorization inclination score, the scores for these questions were averaged ($\alpha$ = .80). Subsequently, the participants completed the manipulation check questions by indicating how much they agreed or disagreed with four statements (e.g., "I need to acquire more resources"; 1 = Strongly disagree, 7 = Strongly agree; Roux et al., 2015). The ratings for these items were averaged ($\alpha$ = .89) to obtain a perceived scarcity score. Finally, the participants were asked to fill in demographic variables, such as gender and age.

## Results

**Manipulation check.**   The manipulation of resource scarcity was successful. Compared to those in the control group ($M_{control}$ = 5.10, $SD_{control}$ = 0.96; t = 3.150, p = .002), the resource scarcity group ($M_{scarcity}$ = 5.51, $SD_{scarcity}$ = 0.90) experienced resource scarcity.

**Moderation analysis.**   To examine the moderating effect of mutability, we conducted a linear regression, with general categorization tendency as the dependent variable and resource scarcity (1 = control, 2 = scarcity), perceived mutability ($M$ = 4.61), and their interaction as independent variables. The results support our hypothesis because a significant interaction effect was observed ($\beta$ = -0.42, $p$ = 0.001). To understand this role further, we used the floodlight analysis technique (Johnson-Neyman technique in SPSS PROCESS; see [45]). Fig 1 shows the results. Using the Johnson-Neyman approach, the lighting analysis showed that at

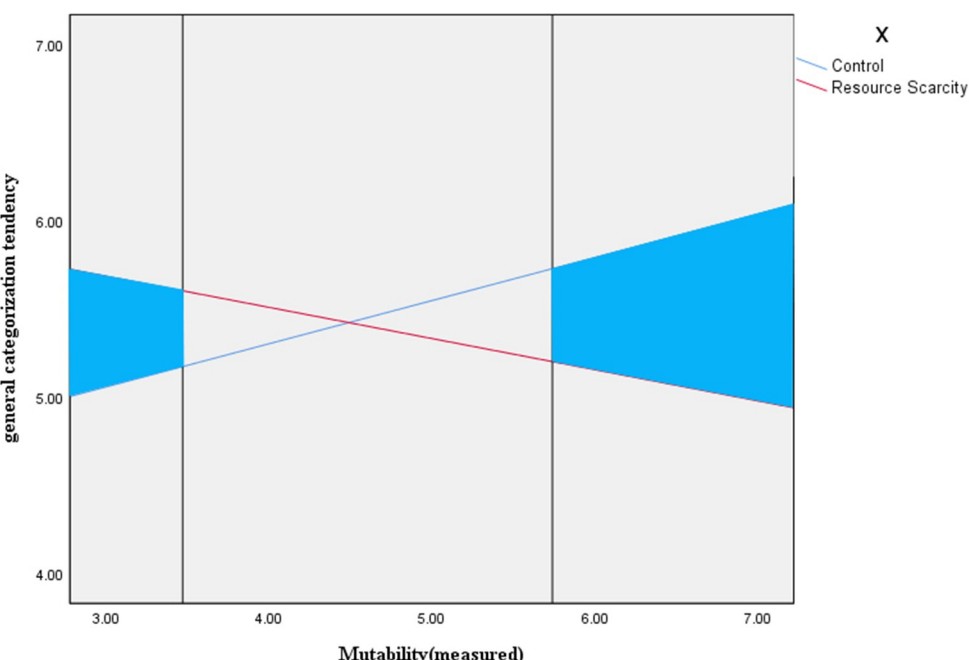

**Fig 1. Floodlight analyses results, effect of resource scarcity on general categorization tendency for different levels of mutability.**

1SD higher than the average of the mutability scale (5.74), compared to the baseline group, the resource-constraint group showed a lower inclination to categorize (B = -.52 with 95% CI of [-.90 -.14], SE = .19, t = -2.73, p = .006), while at 1SD lower than the average of the mutability scale (3.47), compared to the baseline group, the resource-constraint group showed a higher inclination to categorize (B = .42, 95% CI [.04 .80], SE = .19, $t$ = 2.18, $p$ =. 029). Accordingly, H1 stands supported.

### Discussion

Experiment 1 showed that the impact of resource constraints on general categorization tendency depends on the perceived mutability of resource disparity. Under conditions of low mutability, resource constraints increase individuals' general categorization tendency. Conversely, under conditions of high mutability, resource constraints reduce individuals' general categorization tendency. In Experiment 2, we examined the psychological mechanism underlying this effect.

### Study2

The goal of Study 2 was to confirm the results of Study 1 using different resource scarcity manipulation methods and applying a behavioral categorization measure to evaluate general classification inclination. The second goal was to provide further support for the mechanism of the relationship between scarcity and categorical inclination. Specifically, we predicted that the interaction effect of resource constraints and perceived low mutability of scarcity on general categorization tendency will be mediated by the desire for control. In contrast, we also predicted that the desire for abundance would mediate the interactive effect of resource scarcity and perceived high mutability on general categorization tendency.

### Method

**Participants and design.** We recruited 200 participants (57%female; $M_{age}$ = 30.66 years) from the Credemo platform (a small cash incentive was offered). They were randomly allocated to experimental conditions in a 2 (resource scarcity vs. abundance) ×2 (mutability: low vs. high) between-participants design.

**Procedure.** Following the manipulation procedure in the studies by Briers and Laporte (2013) and Yang and Zhang (2022), the participants who were instructed to rate their overall savings on a nine-point scale did the following: those in the resource-scarce group used a scale ranging from one ("RMB 0–5000") to nine ("more than RMB 4,000,000"); those in the resource-rich group used a scale ranging from one to nine ("RMB 0–50" to "more than RMB 400,000"). Participants in both groups composed a short essay describing what it was like to grow up in a poor versus wealthy household.

Thereafter, they performed a reading comprehension task adapted from Kray and Haselhuhn (2007), which manipulated the perceived mutability of resource discrepancy [46]. Under conditions of high mutability, the participants were offered research evidence to support an essay's assertion that individuals may alleviate resource discrepancy through personal efforts. Conversely, participants in the "low mutability condition" read an essay that claimed that, despite reasonable efforts, it is impossible for individuals to alleviate resource discrepancy. Subsequently, the participants were instructed to record one example that supported the essay's main theme. They rated three statements to complete the manipulation check (e.g., "I think I can change resource discrepancy through some efforts"; 1 = "not at all," and 7 = "very much").

The participants then responded to a three-item, seven-point scale to measure the desire for abundance [9]: "I desire to have a lot of things," "I desire to own a lot of things," and "Having a lot of things makes me happy" (1 = "not at all," and 7 = "very much"). The desire for abundance index was determined by averaging the values of these items ($\alpha$ = .95). Thereafter, the participants completed the nine-item Desirability of Control scale [47], which included items, such as "I enjoy having control over my own destiny" and "I enjoy making my own decisions." Subsequently, following Park et al. (2020), we asked the participants to complete a behavioral categorization measure to assess their general categorization tendency. Specifically, a group of four geometric figures—(a) a black circle, (b) a white square, (c) a white circle, and (d) a black square—were presented to individuals, each in two forms and two colors, and the participants responded to four items (e.g., "I would not group object (a) with either (c) or (d)"; 1 = "not at all," and 7 = "very much"). The reasoning behind this was that this situation involves these two most important dimensions for categorization (shape and color) and would be used more frequently by people who have a higher inclination to categorize the given figures. As a result, higher scores on each question indicated a greater general categorization inclination, which was achieved by reverse-coding all items. The scores for these questions were averaged to obtain a general categorization inclination score ($\alpha$ = .81). At the end of the study, the same four questions ($\alpha$ = .89) used in Study One were used to check scarcity manipulation. Finally, the participants responded to basic demographic questions.

## Results

**Manipulation checks.** The manipulation of resource constraints was successful. Compared to the resource-abundance group ($M_{abundance}$ = 5.27, $SD_{abundance}$ = 1.03; $F$ (1, 196) = 7.754, $p$ = .006), the resource-scarcity group ($M_{scarcity}$ = 5.62, $SD_{scarcity}$ = 0.74) experienced a scarcity of resources. Similarly, we averaged scores for the three questions that assessed the manipulation of the mutability of resource scarcity to generate a perceived mutability score. A two-way analysis of variance (ANOVA) validated our manipulation by revealing only one main effect of perceived mutability: participants in the low-mutability group stated that resource scarcity was less mutable ($M$ = 5.42) than participants in the high-mutability group ($M$ = 5.94; $F$ (1, 196) = 15.93, $p < .001$).

**General categorization tendencies.** A 2 (resource scarcity vs. abundance) ×2 (mutability: low vs. high) ANOVA on categorization tendency revealed a significant interaction between scarcity and mutability ($F$ (1, 196) = 13.52, $p < .001$, $\eta^2$ = 0.06). Fig 2 shows the results. Specifically, in the low mutability condition, participants in the resource-constraints group ($M$ = 5.15, $SD$ = 1.04) exhibited much greater general categorization tendency than in the resource-abundance group ($M$ = 4.65, $SD$ = 1.31; $F$ (1, 196) = 3.99, $p$ = 0.04, $\eta^2$ = 0.02). In contrast, in the high-mutability condition, the resource-scarcity group ($M$ = 3.71, $SD$ = 1.11) indicated significantly lower general categorization tendency than the resource abundance group ($M$ = 4.49, $SD$ = 1.39; $F$ (1, 196) = 10.24, $p$ = 0.002, $\eta^2$ = 0.05).

**Mediation analysis.** We examined the moderated mediation model using PROCESS SPSS Macro [48], Model 8, where general categorization tendency was the dependent variable, desire for abundance and desire for control were simultaneous mediators, mutability was the moderator, and resource scarcity was the independent variable. The outcomes confirmed our prediction. More specifically, the desire for abundance reduced individuals' general categorization tendency ($b$ = .28, $t$ (200) = 2.84, $p < 0.001$), and the desire for control enhanced individuals' general categorization tendency ($b$ = -.27, $t$ (200) = -2.58, $p < 0.05$).

There was a significant moderated mediation on the desire for control (score = -.14, SE = 0.089, 95% CI = [-.35, -.01]). For the low-mutability group, the indirect impact of

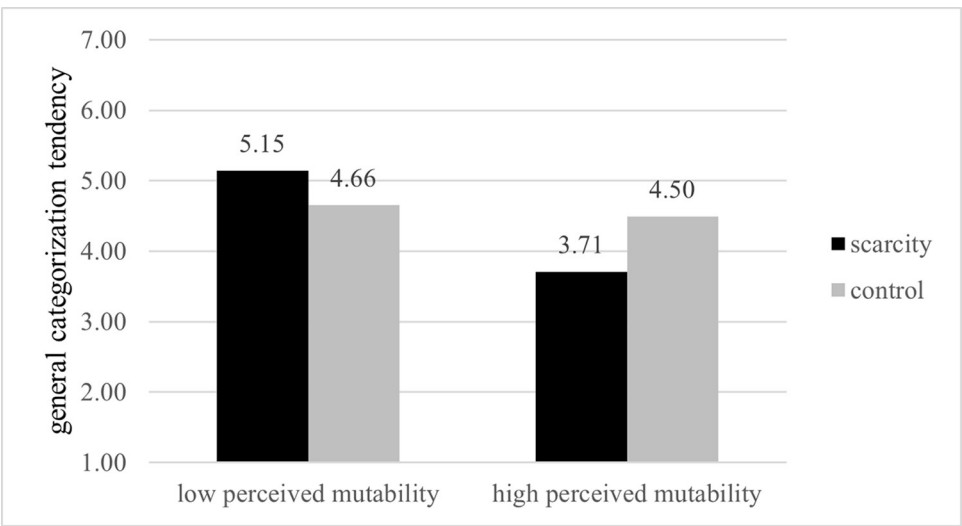

**Fig 2. Interaction effect of resource scarcity and perceived mutability on general categorization tendency.**

resource scarcity on general categorization tendency (via the desire for abundance) was not significant, whereas the indirect impact of resource scarcity on general categorization tendency (via the desire for control) was positive and significant ($b$ = .17, 95% [LLCI, ULCI]) = [.03 .35]). That is, the desire for control-mediated resource scarcity affects the general categorization tendency of individuals with low mutability. Therefore, H2a stands supported.

On the contrary, there was a significant moderation score for the desire for abundance (score = -.15, SE = .11, 95% CI = [-.4295, -.0041]). For the high-mutability group, the indirect impact of resource constraints on general categorization tendency (via the desire for abundance) was negative and significant ($b$ = -.16; 95% [LLCI, ULCI]) = [-.4206 -.0140]), while for the low-mutability group, this was not significant ($b$ = -.0077; 95% [LLCI, ULCI]) = [-.1053 .0859]). That is, the desire for abundance mediated resource scarcity on the general categorization tendency for individuals with high mutability. Therefore, H2b stands supported.

## Discussion

Our hypothesis was further supported by Experiment 2, which replicated the findings of previous studies and demonstrated that the effect of resource scarcity on general categorization tendency depends on the perceived mutability of the resource discrepancy. More importantly, this study supports our proposed mechanism of the desire for control and abundance. Specifically, under the low-mutability condition, resource scarcity stimulates the need for control, which strengthens a person's tendency for general categorization. However, under high-mutability conditions, the desire for abundance is triggered by resource scarcity, which decreases people's general categorization tendency.

## General discussion

### Conclusion

A growing stream of research clearly shows that resource constraints lead to diverse consequences. However, what is less known is the precise nature of these consequences, including when and why they occur. Our research investigates when and why resource constraints sometimes increase individual's general categorization tendency [12, 13, 15] and sometimes

decrease the same [5, 9, 24]. Drawing on two empirical studies, this research demonstrates that the impact of resource scarcity on general categorization tendencies is contingent upon the mutability of resource scarcity. Specifically, we find that individuals are more likely to enhance their general categorization propensity when they perceive resource scarcity as being low in mutability. This effect is driven by their need to re-establish their sense of personal control, which is threatened by the scarcity. In contrast, when resource scarcity is perceived to be high in mutability, individuals' desire for abundance is exacerbated by the scarcity, making categorization particularly unpleasant. As a result, resource scarcity is expected to lower the tendency for categorization in general. These findings contribute to a deeper understanding of how individuals respond to resource scarcity and its implications for their categorization behavior.

## Theoretical contributions

This study makes several theoretical contributions to the existing literature. First, it provides a more detailed explanation of how resource constraints affect general categorization inclination. It offers a solution to reconcile earlier results that appear to be contradictory. Prior research revealed that resource scarcity might augment people's general categorization tendency (e.g. [12]) and can also decrease an individual's general categorization tendency (e.g. [9]). This study showed that individuals use diverse methods to solve the challenge of resource scarcity, which largely depends on the mutability of the resource discrepancy. When people perceive low mutability of the unfavorable resource discrepancy (e.g., a stable cause) and realize that their efforts would not minimize the discrepancy, they will be more inclined toward general categorization tendency. Otherwise, they will exhibit a lower preference for general categorization tendency.

Second, our study showed that resource scarcity triggers two parallel psychological processes that affect an individual's general categorization inclination. Previous literature found that the impact of resource scarcity on categorization only provided inconsistent and limited psychological mechanisms. More specifically, earlier research demonstrated that the desire for control may [10] mediate the impact of resource scarcity on classification tendency, but other studies have relied only on the desire for abundance to explain this relationship [9]. Our work revealed that both the desire for control and abundance are simultaneously counterproductive in the subsequent categorization tendency. Although they were each independently examined in earlier literature, the current study combined them with studies on the perceived mutability of the scarcity scenario and focused on how they both ultimately affect categorization inclination.

Third, this study extended the self-regulatory model of resource scarcity to the downstream consequences pertaining to categorization tendencies. This model has traditionally focused on downstream consequences, such as compensatory consumption [7], planning and setting priorities [19], and creative thinking [23]. Additionally, this scarcity model shows that individuals respond to resource constraints through two different psychological approaches: by directly eliminating differences in resources and gaining control over other areas of their lives [14]. However, there has not been much research on the perceived mutability of the scarcity scenario [7, 8]. The present study bridged this gap by adopting the mutability scales developed by Leary et al. (2022) and adapting manipulation concerning perceived mutability [46]. Thus, we tested the moderating role of mutability on the impact of resource constraints on categorization propensity.

## Practical implications

Our findings provide several managerial implications. Our results suggest that marketers could use flexible strategies for their products or services. In certain situations, we should

increase or decrease consumers' categorization perceptions. For example, product displays (e.g., product grouping labels) can bolster consumers' general categorization tendency. According to our findings, when consumers perceive resource scarcity as being low (high) in mutability, marketers should enhance (reduce) the categorization perceptions of the products or services. In addition, marketers should use the scarcity appeals in proper ways. The effect of scarcity on decision-making is different, since the concept of scarcity is nuanced. We identify the moderating role of mutability. As a result, whether we should activate the perception of resource scarcity depends on the mutability of resource scarcity.

### Limitations and future directions

This study has several limitations that should be addressed in future research. First, considering the internal validity of the results, our study only applied well-developed manipulations. Future studies should examine the comparative prominence and importance of various types of resource scarcity and investigate how they affect people's behavior. Second, the approach is limited by the fact that we assessed individuals' general categorization tendency primarily through self-reports. To further our understanding and increase the external validity of our study, future research should conduct field experiments to test our propositions. Finally, our research is limited to traditional conditions. However, technological innovation has brought changes to our life, especially the usage of AI. Past research indicated that algorithmic consumption was associated with users' cognition and perception [49–51]. Future research could further explore the influence of AI on individuals' general categorization tendency.

## Supporting information

**S1 Dataset. Dataset of Study 1.**
(CSV)

**S2 Dataset. Dataset of Study 2.**
(CSV)

## Author Contributions

**Conceptualization:** Haowen Xiao, Guocheng Li.

**Data curation:** Jiayi Guo, Junyi Fang.

**Funding acquisition:** Haowen Xiao, Ting Xiao.

**Writing – original draft:** Guocheng Li.

**Writing – review & editing:** Ting Xiao.

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
