## [Decision Letter · Decision Letter 0]

27 Mar 2023

PONE-D-23-02056Impact of Resource Scarcity on General Categorization Tendency: The Moderating Role of Perceived MutabilityPLOS ONE

Dear Dr. Xiao,

Thank you for submitting your manuscript to PLOS ONE. After careful consideration, we feel that it has merit but does not fully meet PLOS ONE’s publication criteria as it currently stands. Therefore, we invite you to submit a revised version of the manuscript that addresses the points raised during the review process.

Your manuscript has been reviewed and requires modifications prior to making a decision. The comments of the reviewers are included at the bottom of this letter. Reviewers indicated that introduction, methods and results sections should be improved. We would be glad to consider a substantial revision of your work, where the reviewer’s comments will be carefully addressed one by one.

We look forward to receiving your revised manuscript.

Kind regards,

Asli Suner Karakulah, PhD

Academic Editor

PLOS ONE

2. Please provide additional details regarding participant consent. In the Methods section, please ensure that you have specified (1) whether consent was informed and (2) what type you obtained (for instance, written or verbal). If your study included minors, state whether you obtained consent from parents or guardians. If the need for consent was waived by the ethics committee, please include this information.

“This research was supported by Hainan Province Natural Science Foundation of China (Grant No. 722QN291). Xiao H.W. received this award.”

Reviewers' comments:

Reviewer's Responses to Questions

**Comments to the Author**

1. Is the manuscript technically sound, and do the data support the conclusions?

Reviewer #1: Yes

Reviewer #2: Yes

Reviewer #3: Partly

2. Has the statistical analysis been performed appropriately and rigorously? 

Reviewer #1: Yes

Reviewer #2: N/A

Reviewer #3: I Don't Know

3. Have the authors made all data underlying the findings in their manuscript fully available?

Reviewer #1: Yes

Reviewer #2: Yes

Reviewer #3: No

4. Is the manuscript presented in an intelligible fashion and written in standard English?

Reviewer #1: Yes

Reviewer #2: Yes

Reviewer #3: Yes

5. Review Comments to the Author

Reviewer #1: Good work. I've read it with interest. The paper is well written and nicely thought out. With a minor revision, I believe this paper can be accepted. My only reservation is that a lack of reference. Related research is not well outlined nor considered critically. Thus, I urge the authors to redo literature review. I suggest the following works for your new references:

Hameleers, M., Park, Y., Diakopoulos, N., Helberger, N., Lewis, S., Westlund, O. & Baumann, S. (2022). Countering algorithmic bias and disinformation and effectively harnessing the power of AI in media. Journalism & Mass Communication Quarterly, 99(4), 887-907. https://doi.org/10.1177/10776990221129245

Shin, D. (2023). Embodying algorithms, enactive AI, and the extended cognition: You can see as much as you know about algorithm. Journal of Information Science, 49(1), 18-31. http://doi:10.1177/0165551520985495

Shin, D. (2022). The perception of humanness in conversational journalism: An algorithmic information-processing perspective. New Media & Society, 24(12), 2680-2704. http://doi:10.1177/1461444821993801

Lim, J., Ahmad, N., & Ibarahim, M. (2022). Understanding user sensemaking in fairness and transparency in algorithms: Algorithmic sensemaking in over-the-top platform. AI & Society. https://doi.org/10.1007/s00146-022-01525-9

Reviewer #2: Abstract is well structured and identifies clearly major contributions, although method applied is missing.

Very good read, well structured, and with strong robustness of results that support the study.

Overall, there are small format issues that need to be corrected.

Figure 2 is not mentioned in the document. Variables and parameters of the studies need more explanation.

5 should be Conclusions and 5.1 renamed as Discussion and merged with 4.3

In 5.1 the authors could elaborate more on the results' discussion and clearly relate to the hypothesis formulated in 2.2.

In 5.3 Limitations and Future Directions explain better the difficulties along the study and how the authors have surpassed them and how they plan to tackle them in the future.

Reviewer #3: The paper is generally well written however, the paper is overwhelmingly focused towards theory and lacking context within which the results and findings are applicable. The abstract and the introduction a severely lacking a clear context for the research presented, it needs a clear statement outlining what resources and in what context these are being examined. Further, both figure 1 and 2 were missing from the submission – and thus I was not able to review or consider these as to their accuracy/appropriateness.

I would recommend some further edits as follows:

- Remove “desire for control” and “desire for abundance” from the keywords and replace with ‘self-regulatory model’ and ‘resource discrepancy’ – these would better reflect the content of the paper.

- The opening point in the introduction “Individuals usually perceive resources as scarce” – surely this depends on what the resource is? This is a rather ambiguous statement.

- Also in the introduction on p2 refers to “general categorization tendency” – this needs defining as a concept.

- In section 2.2 “a psychological sense of decrease” – this does not really mean anything – I suggest this is rephrased to improve the clarity of expression.

- In section 2.3 – I am not convinced about the religious reference “those who feel less in control tend to believe more strongly that God is in charge” – can this be explained more from a scientific perspective instead?

- Also in 2.3, “individuals who lack control tend to seek structure (Whitson and Galinsky, 2008), which raises preferences for product logos with bounded (vs. unbounded) and numerical forms of point (vs. range) values” – this statement requires more explanation and definition.

- In 2.4, “caloric foods” – this should be ‘calorific’ or ‘high calorie’.

- At the end of 5.2 “our findings provide an innovative theoretical understanding of coping

- mechanisms that might be employed in response to a generalized feeling of resource constraints” – the authors have not really explored coping mechanisms in the discussion sections per se.

- Finally, in 5.3 – the paper lacks mention of any practical contributions, for instance what does the study conducted show beyond the theory? Are there any managerial implications to consider here? This links back to the lack of practical context of the paper as a whole.

- The final statements regarding conflict of interest etc. appear in numbered sections – 6-9 – I do not believe these should require a numbered section.

6. PLOS authors have the option to publish the peer review history of their article (what does this mean?). If published, this will include your full peer review and any attached files.

Reviewer #1: No

Reviewer #2: **Yes: **Vitória Albuquerque

Reviewer #3: No

---

## [Author Response · Author response to Decision Letter 0]

13 Apr 2023

Dear Editor and reviewers:

Re: “Impact of Resource Scarcity on General Categorization Tendency: The Moderating Role of Perceived Mutability” 

Great thanks for your professional review of our paper. We have benefited greatly from reading your review comments. After careful consideration and study of your comments, we have made extensive corrections to the previous manuscript.

Each comment from all reviewers has been carefully considered and responded to. To make it easier for you to read, we have organized our feedback in the following format. The reviewers' comments are listed in italicized font, and specific concerns have been specifically numbered. Our replies are given in normal font, in addition, corrections and additions to the manuscript are given in red font. 

Below we present our responses to the specific issues raised by each reviewer.

Yours truly,

The authors

 

Response to the journal requirements:

Our response:

Thanks for your reminder, we have changed and rechecked the format to ensure our manuscript meets PLOS’s ONE’s style requirements.

2. Please provide additional details regarding participant consent. In the Methods section, please ensure that you have specified (1) whether consent was informed and (2) what type you obtained (for instance, written or verbal). If your study included minors, state whether you obtained consent from parents or guardians. If the need for consent was waived by the ethics committee, please include this information.

Our response:

We ensure that contest was informed. The participants provided their written informed consent to participate in this study. Additional details were added to the Methods section. Thank you again for your reminder. 

See pp. 461-464

Ethical review and approval were not required for the study on human participants in accordance with the local legislation and institutional requirements. The participants provided their written informed consent to participate in this study.

“This research was supported by Hainan Province Natural Science Foundation of China (Grant No. 722QN291). Xiao H.W. received this award.”

Our response:

We have added the statement in the cover letter. 

Our response:

We have provided the data in the supporting information. 

Our response:

Thank you. We did make sure that corresponding author link his ORCID to the editorial manager account.

Our response:

Thank you for this comment. We have added the ethical concern in the Method section.

See pp. 223-226

Ethical Concern

Ethical review and approval were not required for the study on human participants in accordance with the local legislation and institutional requirements. The participants provided their written informed consent to participate in this study.

Our response:

Thank you for this comment. We have listed the supporting information captions and uploaded the materials.

Our response:

We have checked the references and removed the retracted articles. 

 

Responses to Reviewer # 1 Comments

1. Good work. I've read it with interest. The paper is well written and nicely thought out. With a minor revision, I believe this paper can be accepted. My only reservation is that a lack of reference. Related research is not well outlined nor considered critically. Thus, I urge the authors to redo literature review. I suggest the following works for your new references:

Hameleers, M., Park, Y., Diakopoulos, N., Helberger, N., Lewis, S., Westlund, O. & Baumann, S. (2022). Countering algorithmic bias and disinformation and effectively harnessing the power of AI in media. Journalism & Mass Communication Quarterly, 99(4), 887-907. https://doi.org/10.1177/10776990221129245

Shin, D. (2023). Embodying algorithms, enactive AI, and the extended cognition: You can see as much as you know about algorithm. Journal of Information Science, 49(1), 18-31. http://doi:10.1177/0165551520985495

Shin, D. (2022). The perception of humanness in conversational journalism: An algorithmic information-processing perspective. New Media & Society, 24(12), 2680-2704. http://doi:10.1177/1461444821993801

Lim, J., Ahmad, N., & Ibarahim, M. (2022). Understanding user sensemaking in fairness and transparency in algorithms: Algorithmic sensemaking in over-the-top platform. AI & Society. https://doi.org/10.1007/s00146-022-01525-9

Our response:

We sincerely appreciate the valuable comments. We have checked the literature carefully and added the references into the Limitations and Future Directions section in the revised manuscript. 

(See p.17)

Finally, our research is limited to traditional conditions. However, technological innovation has brought changes to our life, especially the usage of AI. Past researches indicated that algorithmic consumption was associated with users’ cognition and perception [49-52]. Future research could further explore the influence of AI on individuals’ general categorization tendency.

Responses to Reviewer # 2 Comments

1. Abstract is well structured and identifies clearly major contributions, although the method applied is missing. Very good read, well structured, and with strong robustness of results that support the study.

Our response:

Thank you for your excellent comments! Your points are well taken. We have included the methods used in the research in the abstract, where changes could be found in the revised manuscript. 

(See p.1)

Abstract

Prior studies revealed varying effects of resource scarcity on individuals’ general categorization tendency. However, little is known about when and why such differences occur. Based on the self-regulatory model of resource scarcity, we examine whether resource scarcity generates higher or lower general categorization tendency depending on the perceived mutability of the resource discrepancy. We conducted two online experiments to test the hypotheses. The results affirmed that when individuals consider the resource discrepancy to be mutable, they are more likely to seek abundance to compensate for resource scarcity, thus reducing their general categorization tendency. In contrast, perceiving the scarcity as immutable triggers the intention to restore a sense of control undermined by the scarcity, increasing individuals’ general categorization tendency. Our findings provide insights into the downstream consequences of resource scarcity and offer significant managerial implications for coping strategies.

2. Overall, there are small format issues that need to be corrected.

(1) Figure 2 is not mentioned in the document.

(2) 5 should be Conclusions and 5.1 renamed as Discussion and merged with 4.3.

(3) In 5.1 the authors could elaborate more on the results' discussion and clearly relate to the hypothesis formulated in 2.2.

(4) In 5.3 Limitations and Future Directions explain better the difficulties along the study and how the authors have surpassed them and how they plan to tackle them in the future.

Our response:

Thank you for raising these issues. We recognized that there were indeed formatting and writing issues in the original manuscript. Thus, we corrected the formatting issues.

 (1) In the revised manuscript, we have added the missing content. 

(See p.13)

A 2 (resource scarcity vs. abundance) × 2 (mutability: low vs. high) ANOVA on categorization tendency revealed a significant interaction between scarcity and mutability (F (1, 196) = 13.52, p < .001, η2 = 0.06). Figure 2 shows the results. Specifically, in the low mutability condition… 

Fig 2. Interaction effect of resource scarcity and perceived mutability on general categorization tendency

(2) We have renamed 5 into General Discussion. Meanwhile, 5.1 discussion was renamed as conclusion. This section elaborates more on the main conclusion and results' discussion.

(See pp.14-15)

(3) We appreciated the valuable comments. According to the suggestions, we have revised this part and elaborated more on the results' discussion.

(See p.15)

A growing stream of research clearly shows that resource constraints lead to diverse consequences. However, what is less known is the precise nature of these consequences, including when and why they occur. Our research investigates when and why resource constraints sometimes increase individual’s general categorization tendency (Krosch and Amodio, 2014; Van Kerckhove et al., 2020; Yoon and Kim, 2018) and sometimes decrease the same (Briers and Laporte, 2013; Fan et al., 2019; Park et al., 2020). Drawing on two empirical studies, this research demonstrates that the impact of resource scarcity on general categorization tendencies is contingent upon the mutability of resource scarcity. Specifically, we find that individuals are more likely to enhance their general categorization propensity when they perceive resource scarcity as being low in mutability. This effect is driven by their need to re-establish their sense of personal control, which is threatened by the scarcity. In contrast, when resource scarcity is perceived to be high in mutability, individuals' desire for abundance is exacerbated by the scarcity, making categorization particularly unpleasant. As a result, resource scarcity is expected to lower the tendency for categorization in general. These findings contribute to a deeper understanding of how individuals respond to resource scarcity and its implications for their categorization behavior. 

(4) According to the comments, we have further explained the difficulties in the study and supplied the plan to tackle the problem in the future.

(See p.17)

This study has several limitations that should be addressed in future research. First, considering the internal validity of the results, our study only applied well-developed manipulations. Future studies should examine the comparative prominence and importance of various types of resource scarcity and investigate how they affect people’s behavior. Second, the approach is limited by the fact that we assessed individuals’ general categorization tendency primarily through self-reports. To further our understanding and increase the external validity of our study, future research should conduct field experiments to test our propositions. Finally, our research is limited to traditional conditions. However, technological innovation has brought changes to our life, especially the usage of AI. Past research indicated that algorithmic consumption was associated with users’ cognition and perception [49-52]. Future research could further explore the influence of AI on individuals’ general categorization tendency.

Responses to Reviewer # 3 Comments

1. Remove “desire for control” and “desire for abundance” from the keywords and replace with ‘self-regulatory model’ and ‘resource discrepancy’ – these would better reflect the content of the paper.

Our response:

We think it is an excellent suggestion. Thank you for raising this point. Following your suggestion, we have changed the corresponding keywords, which definitely better reflect the content of the paper.

(See p.1)

Keywords: resource scarcity, categorization tendency, perceived mutability, self-regulatory model, resource discrepancy

2. The opening point in the introduction “Individuals usually perceive resources as scarce” – surely this depends on what the resource is? This is a rather ambiguous statement.

Our response:

Thanks for your valuable comments. Considering the possible ambiguity of this statement, we change the sentence to

(See p.1)

 Individuals usually perceive resources, including money (Sharma and Alter, 2012), time (Kapoor and Tripathi, 2020), and product (Zhu and Ratner, 2015) as scarce.

3. Also in the introduction on p2 refers to “general categorization tendency” – this needs defining as a concept.

Our response: 

Based on your suggestions, the required definition and explanations have been added to explain the vague concept. 

 (See p.1)

Recent studies examined how the general sense of resource scarcity influences individuals’ general categorization tendency, which refers to a human’s basic motive to seek structures and order by the process of dividing the world into manageable groups of entities (Park et al., 2020; Van Kerckhove et al., 2020; Yoon and Kim, 2018).

4. In section 2.2 “a psychological sense of decrease” – this does not really mean anything – I suggest this is rephrased to improve the clarity of expression.

Our response: 

Based on your suggestion, we have replaced the phrase with “a feeling of reduction”.

 (See p.5)

This is because categorization evokes a feeling of reduction (i.e., many items are divided into smaller subgroups by categorization).

5. In section 2.3 – I am not convinced about the religious reference “those who feel less in control tend to believe more strongly that God is in charge” – can this be explained more from a scientific perspective instead?

Our response: 

We have rephrased the paragraph and explained more from a scientific perspective.

 (See p.6)

A lack of personal control engenders an aversive feeling that drives individuals to regain their perception of control to baseline levels (Chen et al., 2017). Previous research demonstrated that people who lack personal control are likely to affirm structured interpretations of the world (Landau et al., 2015; Whitson and Galinsky, 2008). For instance, research showed that low personal control led to increased subsequent beliefs in the existence of a controlling God (Laurin et al., 2008). 

6. Also in 2.3, “individuals who lack control tend to seek structure (Whitson and Galinsky, 2008), which raises preferences for product logos with bounded (vs. unbounded) and numerical forms of point (vs. range) values” – this statement requires more explanation and definition.

Our response: 

We have rephrased the paragraph and explained more from a scientific perspective.

(See p.6)

In marketing domain, consumers who lack control tend to seek structure through consumption. Lembregts and Pandelaere (2019) show that low personal control leads consumers to prefer and rely more on numerical attributes as a point value, relative to a range format. This occurs because product attributes specified in a point value format strengthen consumers’ belief that the environment is predictable.

7. In 2.4, “caloric foods” – this should be ‘calorific’ or ‘high calorie’.

Our response: 

We sincerely thank your careful reading. We feel sorry for our mistakes. As suggested by the reviewer, we have corrected the “caloric foods” into “calorific ”

 (See p.7)

Similarly, Kapoor and Tripathi (2020) suggested that people who experience downward timekeeping are more likely to perceive themselves as resource-scarce, leading to a higher preference for high calorie foods.

8. At the end of 5.2 “our findings provide an innovative theoretical understanding of coping- mechanisms that might be employed in response to a generalized feeling of resource constraints” – the authors have not really explored coping mechanisms in the discussion sections per se.

Our response: 

After serious consideration of your valuable comments and careful examination of our paper, we have recognized that there is really no coping mechanism explored in this section. As a result, we have removed this sentence. Changes can be founded in the revised manuscript. 

9. Finally, in 5.3 – the paper lacks mention of any practical contributions, for instance what does the study conducted show beyond the theory? Are there any managerial implications to consider here? This links back to the lack of practical context of the paper as a whole.

Our response: 

Thank you for this comment! We have added the practical implications section.

(See p.16)

Practical implications

Our findings provide several managerial implications. Our results suggest that marketers could use flexible strategies for their products or services. In certain situations, we should increase or decrease consumers’ categorization perceptions. For example, product displays (e.g., product grouping labels) can bolster consumers’ general categorization tendency. According to our findings, when consumers perceive resource scarcity as being low (high) in mutability, marketers should enhance (reduce) the categorization perceptions of the products or services. In addition, marketers should use the scarcity appeals in proper ways. The effect of scarcity on decision-making is different, since the concept of scarcity is nuanced. We identify the moderating role of mutability. As a result, whether we should activate the perception of resource scarcity depends on the mutability of resource scarcity.

10. The final statements regarding conflict of interest etc. appear in numbered sections – 6-9 – I do not believe these should require a numbered section.

Our response: 

Thanks for your comments. Based on your suggestion, we have removed these numbers to optimize the structure of the paper. Traces of alternations can be found in the resubmitted manuscript clearly. 

Based on your valuable comments, we have tried our best to improve the manuscript and made some changes marked in red in revised paper which will not influence the content and framework of the paper. We appreciate for your serious work, and hope correction will meet with approval. Once again, thanks for your suggestions. 

Yours truly,

The authors

---

## [Decision Letter · Decision Letter 1]

15 May 2023

PONE-D-23-02056R1Impact of Resource Scarcity on General Categorization Tendency: The Moderating Role of Perceived MutabilityPLOS ONE

Dear Dr. Xiao,

Thank you for submitting your manuscript to PLOS ONE. After careful consideration, we feel that it has merit but does not fully meet PLOS ONE’s publication criteria as it currently stands. Therefore, we invite you to submit a revised version of the manuscript that addresses the points raised during the review process.

Thank you for revising your manuscript. However, there are still a few issues that need to be addressed. Please respond to the Reviewer 3’s comments seriously. I invite you to respond to the comments and revise your manuscript.

We look forward to receiving your revised manuscript.

Kind regards,

Asli Suner Karakulah, PhD

Academic Editor

PLOS ONE

Journal Requirements:

Reviewers' comments:

Reviewer's Responses to Questions

**Comments to the Author**

1. If the authors have adequately addressed your comments raised in a previous round of review and you feel that this manuscript is now acceptable for publication, you may indicate that here to bypass the “Comments to the Author” section, enter your conflict of interest statement in the “Confidential to Editor” section, and submit your "Accept" recommendation.

Reviewer #1: All comments have been addressed

Reviewer #2: All comments have been addressed

Reviewer #3: (No Response)

2. Is the manuscript technically sound, and do the data support the conclusions?

Reviewer #1: Yes

Reviewer #2: Yes

Reviewer #3: Yes

3. Has the statistical analysis been performed appropriately and rigorously? 

Reviewer #1: Yes

Reviewer #2: I Don't Know

Reviewer #3: Yes

4. Have the authors made all data underlying the findings in their manuscript fully available?

Reviewer #1: Yes

Reviewer #2: (No Response)

Reviewer #3: No

5. Is the manuscript presented in an intelligible fashion and written in standard English?

Reviewer #1: Yes

Reviewer #2: Yes

Reviewer #3: Yes

6. Review Comments to the Author

Reviewer #1: Thank you for submitting your revision. I found the revised work well done and now I can safely accept the revision for publication.

The manuscript has been greatly improved in terms of content and structure. The findings will be important theoretical advancement as well as practical implications.

The results affirmed that when individuals consider the resource discrepancy to be

mutable, they are more likely to seek abundance to compensate for resource

scarcity, thus reducing their general categorization tendency. In contrast, perceiving

the scarcity as immutable triggers the intention to restore a sense of control

undermined by the scarcity, increasing individuals’ general categorization tendency.

The findings provide insights into the downstream consequences of resource scarcity

and offer significant managerial implications.

Great work.

Reviewer #2: Excellent review of the document, tackling all comments and adding new features that enriched the study.

Reviewer #3: Thank you for addressing my suggestions and comments on your article. In my opinion the revised and amended version is significantly improved. The addition of the practical recommendations section is helpful too.

Since I had not previously had sight of figures 1 and 2 until the revised manuscript had been submitted I have just one minor point – figure 2 is missing a label on the y-axis – what is the 1.00 to 7.00 scale measuring?

7. PLOS authors have the option to publish the peer review history of their article (what does this mean?). If published, this will include your full peer review and any attached files.

Reviewer #1: No

Reviewer #2: No

Reviewer #3: No

---

## [Author Response · Author response to Decision Letter 1]

16 May 2023

Author’s Response to Reviewers

Dear Editor:

Re: “Impact of Resource Scarcity on General Categorization Tendency: The Moderating Role of Perceived Mutability” (Manuscript: PONE-D-23-02056)

On behalf of all the contributing authors, I would like to express our sincere appreciation for your letter and the reviewers’ comments on our paper. 

Each comment from all reviewers has been carefully considered and responded to. To make it easier for you to read, we have organized our feedback in the following format. The reviewers' comments are listed in italicized font, and specific concerns have been specifically numbered. Our replies are given in normal font, in addition, corrections and additions to the manuscript are given in red font. 

Thank you again for taking the time to carefully review our manuscript. We are grateful to have you as the editor of our paper.

Below we present our responses to the specific issues raised by each reviewer.

Yours truly,

The authors

Response to the review 3:

1. Since I had not previously had sight of figures 1 and 2 until the revised manuscript had been submitted I have just one minor point – figure 2 is missing a label on the y-axis – what is the 1.00 to 7.00 scale measuring?

Our response:

Thanks for your careful review and reminder, we have rechecked and added the label on the y-axis of Figure 2 to exactly express the meaning of the 1.00 to 7.00 scale measuring. Changes could be found clearly in our resubmitted manuscript.

 (See p.13 and line.309)

Fig 2. Interaction effect of resource scarcity and perceived mutability on general categorization tendency.

---

## [Decision Letter · Decision Letter 2]

22 May 2023

Impact of Resource Scarcity on General Categorization Tendency: The Moderating Role of Perceived Mutability

PONE-D-23-02056R2

Dear Dr. Xiao,

We’re pleased to inform you that your manuscript has been judged scientifically suitable for publication and will be formally accepted for publication once it meets all outstanding technical requirements.

Kind regards,

Asli Suner Karakulah, PhD

Academic Editor

PLOS ONE

Additional Editor Comments (optional):

The authors addressed the reviewers' concerns and substantially improved the content of MS.

So, based on my own assessment as an academic editor, MS can be accepted in its current form.

Reviewers' comments:

Reviewer's Responses to Questions

**Comments to the Author**

1. If the authors have adequately addressed your comments raised in a previous round of review and you feel that this manuscript is now acceptable for publication, you may indicate that here to bypass the “Comments to the Author” section, enter your conflict of interest statement in the “Confidential to Editor” section, and submit your "Accept" recommendation.

Reviewer #3: All comments have been addressed

2. Is the manuscript technically sound, and do the data support the conclusions?

Reviewer #3: (No Response)

3. Has the statistical analysis been performed appropriately and rigorously? 

Reviewer #3: (No Response)

4. Have the authors made all data underlying the findings in their manuscript fully available?

Reviewer #3: (No Response)

5. Is the manuscript presented in an intelligible fashion and written in standard English?

Reviewer #3: (No Response)

6. Review Comments to the Author

Reviewer #3: (No Response)

7. PLOS authors have the option to publish the peer review history of their article (what does this mean?). If published, this will include your full peer review and any attached files.

Reviewer #3: No

---

## [Editor Report · Acceptance letter]

11 Aug 2023

PONE-D-23-02056R2 

Impact of Resource Scarcity on General Categorization Tendency: The Moderating Role of Perceived Mutability 

Dear Dr. Xiao:

I'm pleased to inform you that your manuscript has been deemed suitable for publication in PLOS ONE. Congratulations! Your manuscript is now with our production department. 

Kind regards, 

on behalf of

Dr. Asli Suner Karakulah 

Academic Editor

PLOS ONE